# Molecular Insights into the Centaurea Calocephala Complex (Compositae) from the Balkans—Does Phylogeny Match Systematics?

Jelica Novaković [1,*], Pedja Janaćković [1], Alfonso Susanna [2], Maja Lazarević [1], Igor Boršić [3], Sretco Milanovici [4], Dmitar Lakušić [1], Bojan Zlatković [5], Petar D. Marin [1] and Núria Garcia-Jacas [2]

1    Institute of Botany and Botanical Garden "Jevremovac", Faculty of Biology, University of Belgrade, Studentski trg 16, 11000 Belgrade, Serbia; pjanackovic@bio.bg.ac.rs (P.J.); majat@bio.bg.ac.rs (M.L.); dlakusic@bio.bg.ac.rs (D.L.); pdmarin@bio.bg.ac.rs (P.D.M.)
2    Institut Botànic de Barcelona (IBB, CSIC-Ajuntament de Barcelona), Passeig del Migdia s. n., Parc de Montjuïc, 08038 Barcelona, Spain; asusanna@ibb.csic.es (A.S.); ngarciajacas@ibb.csic.es (N.G.-J.)
3    Ministry of Economy and Sustainable Development, Radnička cesta 80/7, 10000 Zagreb, Croatia; igor.borsic@gmail.com
4    Nature Science Department, National Museum of Banat, Huniade Square 1, 300002 Timisoara, Romania; orchids_mils@yahoo.com
5    Department of Biology and Ecology, Faculty of Sciences and Mathematics, University of Niš, Višegradska 33, 18000 Niš, Serbia; bojanzlat@yahoo.com
*    Correspondence: jelica@bio.bg.ac.rs

**Abstract:** Groups of recent speciation are characterized by high levels of introgression and gene flow, which often confounds delimitation of species on a DNA basis. We analyzed nuclear DNA sequences (ETS spacer and the *AGT1* gene) obtained from a large sample of the *C. calocephala* complex from the Balkan clade of *Centaurea* sect. *Acrocentron* (Compositae, Cardueae-Centaureinae) together with a wide representation of other species from the section. Our main goals were to verify the monophyly of the complex as currently defined and to examine the possible presence of introgression and gene flow. Within the complex, species are well-delimited from a morphological point of view and probably originated by allopatric speciation in the Balkan Peninsula. Our results confirm that the Balkan–Eurasian complex is a natural group, but the Centaurea calocephala complex shows a very complicated pattern and its phylogeny is not resolved. Our hypothesis suggests that altitudinal shifts in the transits from glacial to interglacial periods caused successive hybridization events, which are very evident from the DNA networks, between taxa not currently sympatric. As a result, confirmation of interspecific boundaries using molecular markers is extremely complicated.

**Keywords:** *Acrocentron*; *AGT1*; Balkan refugium; *Centaurea*; chromosomes; ETS; gene flow; introgression; low copy genes; species boundaries



## 1. Introduction

The Balkan Peninsula is located in the Southeast of Europe and harbors a very rich flora with more than 7500 species of native plants [1,2], approximately one-third of them endemic [3]. The biodiversity of this region is the result of a complex geological history and interactions between populations, species, and ecosystems [4]. The Balkans were one of the major glacial refugia for plants of the temperate zone, whence a post-glacial expansion to the north took place [5–10]. Many molecular and phylogeographic studies have been conducted to clarify genetic relationships between taxa from this area, e.g., *Campanula* [11–17], *Tanacetum* [18], *Arundo* [19], *Cardamine* [20–22], *Cerastium* [23,24], *Edraianthus* [25–28], *Cyanus* [29], *Salvia* [29,30], *Goniolimon* [31], *Sesleria* [32,33], and *Veronica* [34].

After Quaternary glaciations, rapid radiations occurred within genera with generally herbaceous representatives [35], and a good example is the genus *Centaurea* L. *Centaurea*

belongs to subtribe Centaureinae (Cass.) Dumort., tribe Cardueae Cass. from the family Compositae Giseke, and comprises more than 250 species, divided into 40 sections [36]. *Centaurea* sect. *Acrocentron* (Cass.) DC., one of the largest sections of the genus, is widely distributed, mainly in the Mediterranean region, and includes around 100 species [37–39]. This section is well-defined in terms of the following characters: very large capitula; peripheral florets sterile, without staminodes but provided with achenioids; spiny, long decurrent appendages; and presence of the "Centaurea scabiosa" pollen type [40,41]. The basic chromosome numbers in sect. *Acrocentron* are $x$ = 10 (more frequent) and $x$ = 11 [42–45]. According to the phylogeny by [46], species of sect. *Acrocentron* constitute a natural group and they are classified in several well-supported, monophyletic clades: Balkan–Eurasian, Iberian, North African, Aegean, and Anatolian–Iranian clade.

Within sect. *Acrocentron*, there are some groups of taxa with complex taxonomy. One of them is the Centaurea calocephala complex from the Balkans. Species from the Centaurea calocephala complex are characterized by large, showy capitula, and coriaceous, ovate to oblong involucral bracts with large, shortly decurrent appendages usually covering the bracts; mucronulate to shortly spinose at apex; yellow, brown, or black in the central part; and fimbriate at margins [47,48]. The typical species of this complex are *Centaurea calocephala* Willd. (syn. *Centaurea atropurpurea* Waldst. and Kit.).; *C. chrysolepis* Vis.; *C. crnogorica* Rohlena; *C. gjurasinii* Bošnjak; *C. grbavacensis* (Rohlena) Stoj. and Acht.; *C. immanuelis-loewii* Degen; *C. kotschyana* Heuffel ex Koch; *C. melanocephala* Pančić; *C. murbeckii* Hayek; *C. orientalis* L.; and *C. zlatiborensis* Zlatković, Novaković and Janaćković. All species are morphologically well defined (Figures 1 and 2), and although their geographical distribution is partly overlapping in the big picture, populations generally have an allopatric distribution. It is extremely rare for different species of this complex to coexist (syntopia).

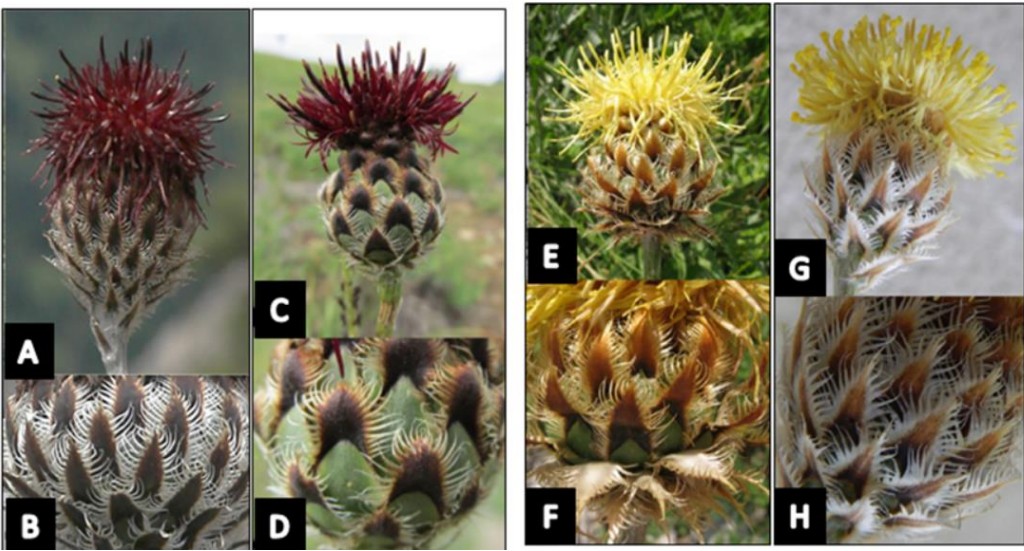

**Figure 1.** Comparison of representatives of the Centaurea calocephala complex from the Balkans (**A**,**B**) *Centaurea calocephala*; (**C**,**D**) *Centaurea zlatiborensis*; (**E**,**F**) *Centaurea gjurasinii*; (**G**,**H**) *Centaurea crnogorica*. Photos: (**A**,**B**) Bogdan Hurdu; (**C**,**D**,**G**,**H**) Pedja Janaćković; (**E**,**F**) Dmitar Lakušić.

Many species of this complex are distributed in different parts of the Balkan Peninsula, and some of them have a wider distribution than the rest: *C. calocephala* is a Balkan-Carpathian subendemic, *C. kotschyana* is a Balkan–Carpathian and Central Europe mountain plant, and *C. orientalis* has a Pontic–Submediterranean distribution (Table S1). The ecological optimum of all species is found in different types of grass communities, mostly on carbonated soils on limestone, dolomite, marble, or loess, less often on ultramafic or siliceous rocks. They usually grow in dry rocky grasslands, rarely on screes, vertical cliffs, loess steppic grasslands, or mesophilous mountain and subalpine meadows and pastures (Table S1).

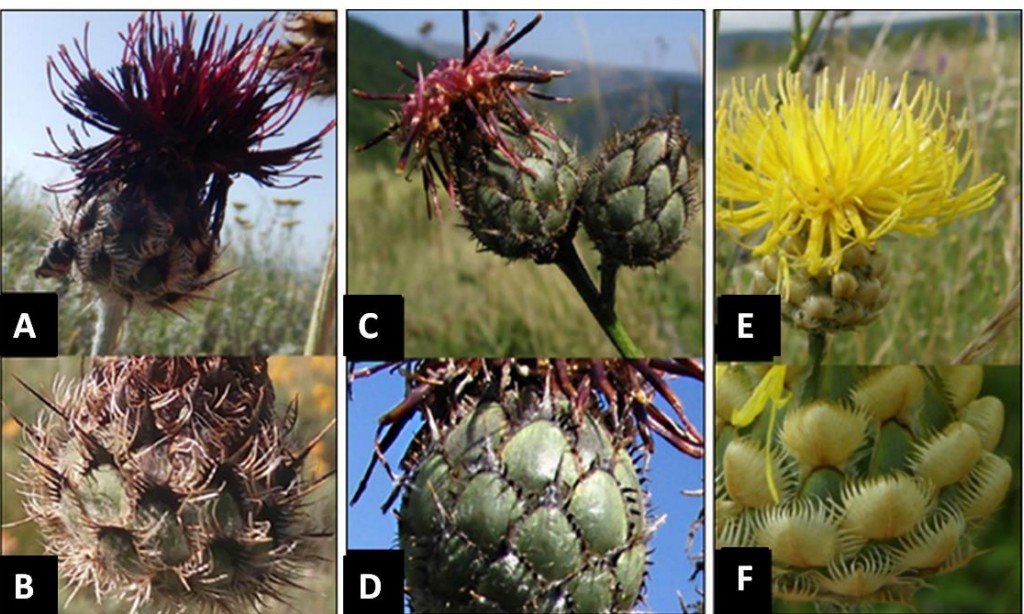

**Figure 2.** Comparison of representatives of the Centaurea calocephala complex from the Balkans (continued). (**A**,**B**) *Centaurea grbavacensis*; (**C**,**D**) *Centaurea melanocephala*; (**E**,**F**) *Centaurea orientalis*. Photos: (**A**,**B**) Jelica Novaković; (**C**,**D**) Bojan Zlatković; (**E**,**F**) Pedja Janaćković.

One of the main drivers of speciation in *Centaurea* is hybridization [42,49–52]. Hybridization is often associated with the emergence of new species, with or without changes in chromosome number [53,54]. Homoploid hybrid species arise when there is no change in ploidy. Homoploid speciation presents additional challenges in relation to allopolyploid speciation, because chromosome doubling prevents backcrossing with parent species, and the chromosome set of the new species contains complete parental genomes [55]. In the absence of polyploidy, incompatibility between parental genomes may cause unsustainability of the homoploid hybrid, except when hybridization occurs between closely related taxa [56,57]. In addition, the lack of reproductive, ecological, and spatial barriers between new homoploid hybrids and their ancestors often leads to introgression and genetic intertwining with their parents, preventing hybrid persistence and their subsequent stabilization and speciation [55,58]. Therefore, known homoploid hybrids are rarer than polyploids. However, recent molecular analyses have revealed a number of homoploid hybrids, and surely the frequency of homoploid hybridization has been miscalculated because it is not as obvious as allopolyploidy [59,60]. Homoploid hybridization is common between related *Centaurea* species [60,61]. The Balkan Centaurea calocephala complex is probably not an exception, as intermediate individuals between some species have been detected [47] and hybridization is usually homoploid because most of species are diploid (Table S2). Dispersal of pollen and achenes in *Centaurea* is restricted and hybridization is possible only in the same geographical area where two different lineages meet [62,63]. Therefore, historical changes in the area of distribution of the species are usually invoked to explain gene flow between taxa not sympatric at present [51].

Molecular analyses can detect hybridization as well as identify parental species [64], and they may reveal events that are not evident from morphology at present, for example, old hybridization events between species [51]. Low-copy genes (e.g., *AGT1*) are excellent for detecting the parental species of hybrids because they have high rates of evolution and are present in the genome with one to few copies [64,65]. Nuclear ribosomal ETS can also be used in cases of introgression and reticulation because concerted evolution is generally incomplete in *Centaurea* sect. *Acrocentron* and different copies often persist [46,51,64]. Moreover, there are other examples of several copies in nuclear-ribosomal DNA in diploids: *Picris* [66], *Rheum* (ITS, cf. [67]), or *Leucanthemum* (ETS, cf. [68]). We gathered a large sample

of the *C. calocephala* complex and sequenced ETS region and the *AGT1* gene with the aims to (a) test the monophyly of the Balkan clade of sect. *Acrocentron* on a much wider sampling; (b) check the monophyly of the *C. calocephala* complex as currently defined; and (c) examine the possible presence of introgression and gene flow in the complex.

## 2. Materials and Methods

### 2.1. Plant Material

We included all species of *Centaurea* sect. *Acrocentron* from the Balkans, following the classification by [47]. Sampling focused on *C. calocephala*, with 10 populations (Figure 3), together with 12 populations of 9 species from the *C. calocephala* complex. From each population, we included one individual, with the exception of three mixed populations from Đerdap, Lika, and Suva Gora, for which we used from two to three individuals. To verify whether the Balkan group of species is monophyletic, we included a very wide representation of species of *Acrocentron* from the rest of the area of the section: Iran, Armenia, Turkey, Greece and the Aegean Islands, the Italian Peninsula and Sicily, North Africa, and the Iberian Peninsula (46 species). The ETS sequences of most of the non-Balkan species were taken from [46]. In total, the study comprised 72 populations of 61 species. The localities, populations, and GenBank accession numbers are detailed in Table S2.

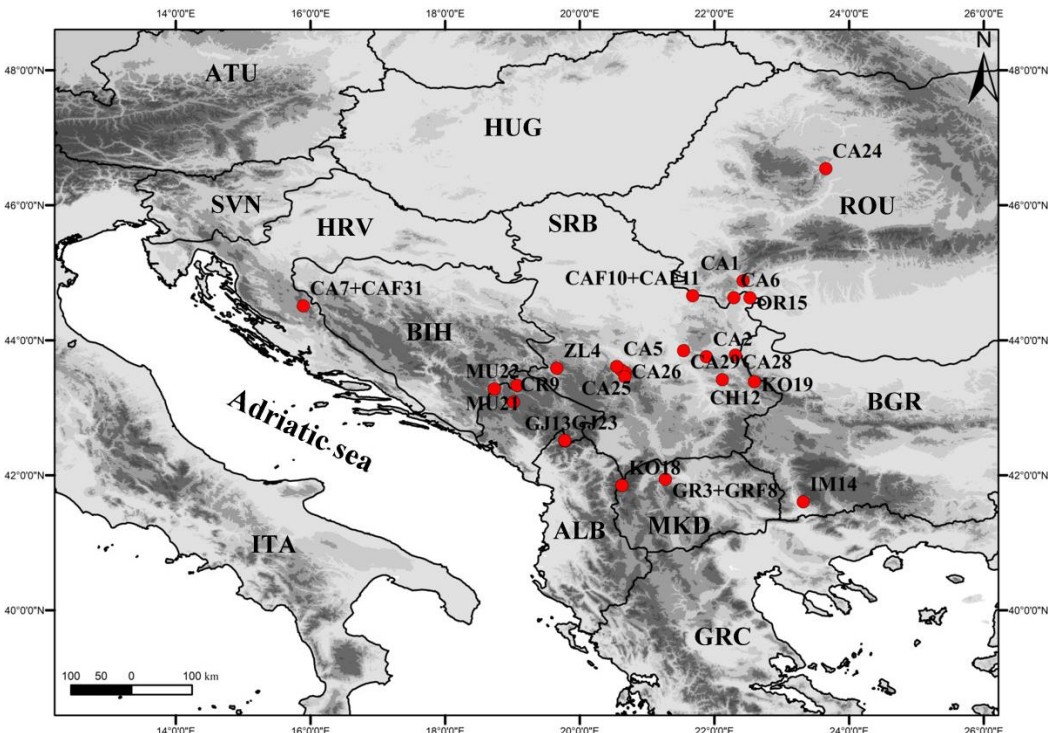

**Figure 3.** Distribution map of the studied populations of the *C. calocephala* complex. Country acronyms: ALB—Albania, AUT—Austria, BIH—Bosnia and Hercegovina, BGR—Bulgaria, HRV—Croatia, GRC—Greece, HUN—Hungary, ITA—Italy, MKD—North Macedonia, MNE—Montenegro, ROU—Romania, SRB—Serbia, SVN—Slovenia. Population codes are detailed in Table S2.

### 2.2. DNA Extraction, Amplification, Cloning, and Sequencing

Genomic DNA was extracted from silica-dried leaves collected in the field using the CTAB protocol described by [69] as modified by [70,71].

The ETS region was amplified with the forward primer ETS1F [72] and the reverse primer 18S-ETS [73]. The profile and reactions used for PCR amplification were the same as described in [74]. The *AGT1* gene was amplified using the specific forward primer Agt1F-podos previously designed by [75] and the universal reverse primer Agt1R [76]. The PCR reaction and amplification profiles were as described in [75].

All PCR products of the ETS region and the *AGT1* gene from most of Balkan populations were cloned (Table S2) using a TOPO TA Cloning® Kit (Invitrogen, Carlsbad, CA, USA). Whenever possible, 8 to 16 positive colonies from each PCR reaction were screened with direct PCR using universal primers T7 and M13 following the protocol of [77]. Eight to ten PCR products were selected for sequencing using the same universal primers previously mentioned.

The PCR products were purified using a QIAquick PCR Purification Kit (Qiagen Inc., Valencia, CA, USA). Sequencing was performed on an ABI 3730xl (Applied Biosystems) following the manufacturer's protocol at Macrogen Inc., Korea.

### 2.3. Phylogenetic Analyses

Sequences were aligned visually using BioEdit 7.0.5.3 [77]. Single substitutions in clones from a single accession were excluded. Consensus sequences were generated for some accessions and regions, condensing single base-pair differences between clones. This reduced the size of the matrices as well as the impact of PCR artifacts (chimeric sequences and Taq errors; cf. [78,79]). For verifying the presence of possible recombinant sequences, datasets were checked using RDP4.97β [80].

We used three datasets. Dataset 1 comprised 88 sequences of the ETS region from a representative sampling of sect. *Acrocentron* from all its distribution area following [46]. Dataset 2 included 66 cloned sequences of the ETS region of populations from the Calocephala complex only. Dataset 3 included 41 sequences of populations of the Calocephala complex of cloned *AGT1* gene. On dataset 1, we carried out a Bayesian inference analysis using the evolutionary model determined by jModeltest v.2.1.10 [81]. The model GTR + G was selected as the best-fit model of nucleotide substitution using Akaike information criteria (AIC). Bayesian inference analysis was carried out using Mr. Bayes v. 3.2 [82] using default prior settings. Analysis was initiated with random starting trees, and four Markov chains were run simultaneously for $30 \times 10^6$ generations. We saved one out every 1000 generations, and the first 7500 generations were discarded as the "burn-in" period after confirming that log-likelihood values had stabilized. Posterior probabilities $\geq 0.95$ were considered statistically significant.

On datasets 2 and 3, we carried out distance network analyses (split graphs) in order to represent groupings in the data and evolutionary distance between pairs of taxa simultaneously. We used the Neighbor-Net (NN) algorithm [83] as implemented in SplitsTree4 v4.13.1 software [84] with the criterion set to uncorrected pairwise (p) distances and including gaps.

### 2.4. Chromosome Number

Chromosome numbers were determined in 18 populations of 10 taxa (Table S2). For the examination of mitotic chromosomes, we excised root tips from germinating seeds and pre-treated them with 0.002 M 8-hydroxyquinoline for 4 h at 8 °C. Pretreated tips were fixed in cold 3/1 (*v/v*) absolute ethanol/glacial acetic acid for 48 h and stored in 70% ethanol at 4 °C for further use. They were hydrolyzed in 1N HCl for 11 min at 60 °C and stained in Schiff's reagent [85] for at least 2 h and squashed in a drop of acetic carmine. Metaphasic plates were examined and photographed using a Leica DMLS light microscope equipped with a Leica DCF 295 digital camera. Chromosome numbers were verified at least on five germinated seeds from five individuals and at least 10 cells per root tip.

## 3. Results

### 3.1. Phylogenetic Analysis

The majority-rule consensus resulting from the Bayesian analysis of dataset 1 (ETS region) is illustrated in Figure 4. The outline of the main geographical clades suggested by [46] is confirmed: there is a Balkan and Eurosiberian clade; an Iberian clade, including one species (*C. xaveri*) from Africa; a North African clade including some Iberian species, sister to an Aegean clade; and an Anatolian–Iranian clade sister to the rest of clades

(Figure 4). The monophyly of the Balkan clade is also well supported. This group of taxa is more related to the Eurasian species than to the Aegean and Anatolian taxa because the representatives of the Eurasian species, namely, *C. cephalariifolia*, *C. glehnii*, *C. legionis-septimae*, and *C. scabiosa*, are nested in the clade. This inclusion, together with the addition of Balkan *C. jankae*, makes the *C. calocephala* complex sensu stricto not monophyletic.

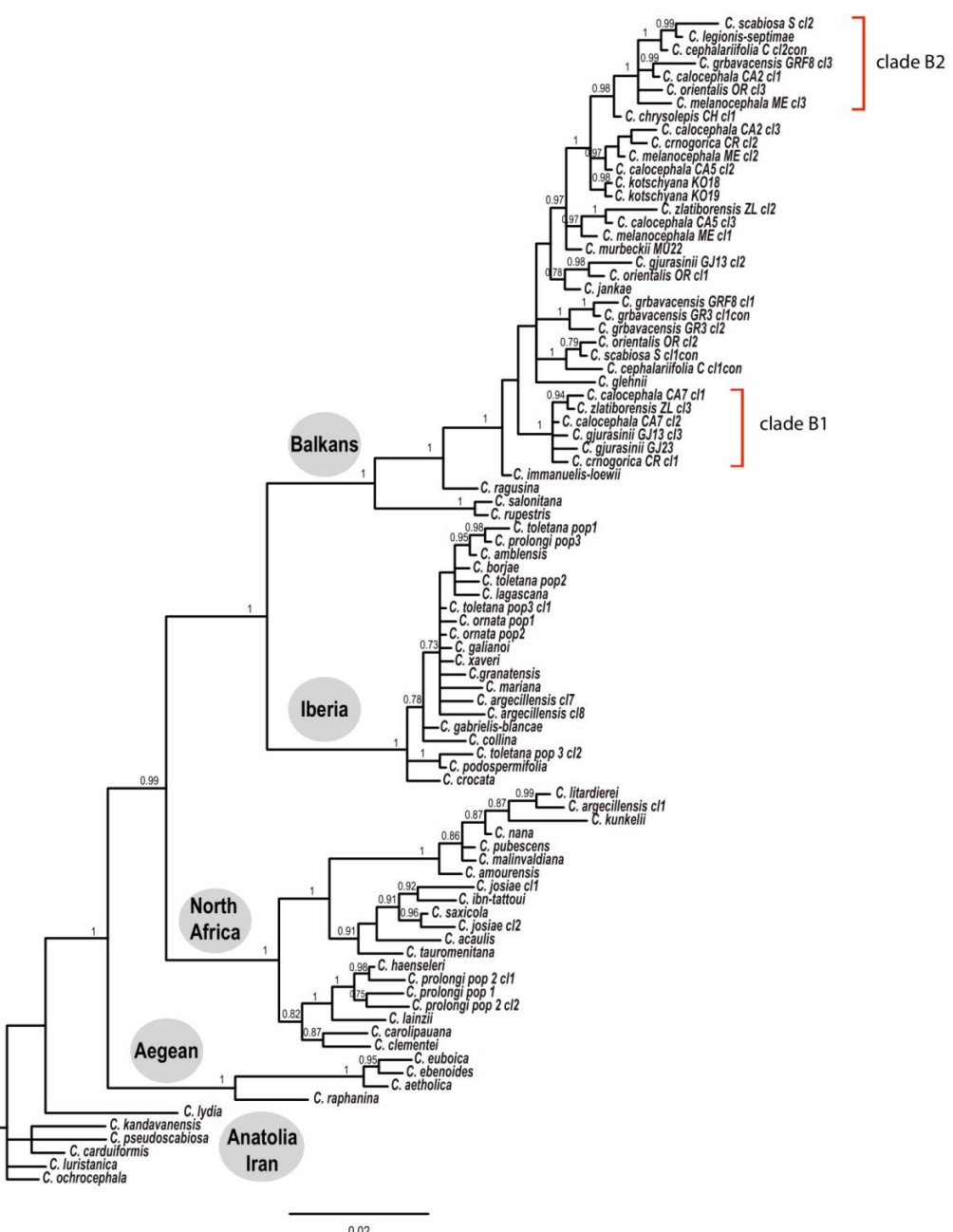

**Figure 4.** Consensus phylogram obtained from 9000 Bayesian trees with higher posterior probability (PP) from ETS data with the Anatolian–Iranian clade as an outgroup. Population codes are detailed in Table S2. Abbreviations: cl = clon; Pop = population; con = consensus sequence.

Within the Balkan clade (incl. the *C. calocephala* complex), the sister species to the rest are successively the clade of *C. salonitana* and *C. rupestris*, two widely distributed species in the region and also in Greece (*C. salonitana* has a very wide distribution reaching from Turkey to the Balkans); then, *C. ragusina*, a narrow endemic from Croatia. The tree also shows that *C. calocephala* has several different ribotypes.

*3.2. Network Analyses*

After checking the presence of recombinant sequences using RDP software [80] and discarding only one sequence for the *AGT1* dataset, we built two networks. The high number of ties detected in the analyses of the two regions agrees with previous evidence of the intense hybridization and introgression in sect. *Acrocentron*, a fact that confounds any interpretation of networks. In fact, almost none of the species with more than one population in the analysis was distributed in a single group in the networks (Figures 5 and 6). Despite the fact that the results are difficult to interpret, two groups could be defined in the ETS network. The first one is the Calocephala 1 group (Figure 5), including clones from several populations of *C. calocephala* (CA1, CA2, CA5, CA6, CA24, CA25, CA26, CA28, and CA29), then *C. calocephala* var. *mixta* (CAF11), *C. chrysolepis* (CH), *C. crnogorica* (CR), *C. grbavacensis* f. *lutea* (GRF8), *C. kotschyana* (KO18 and KO19), *C. melanocephala* (ME), and *C. orientalis* (OR). The Calocephala 2 group includes clones of *C. calocephala* (CA1, CA5, CA6, CA7, CA24, CA26, and CA29), *C. calocephala* var. *flava* (CAF10 and CAF31), *C. calocephala* var. *mixta* (CAF11), *C. chrysolepis* (CH), *C. crnogorica* (CR), *C. gjurasinii* (GJ13 and GJ23), *C. grbavacensis* (GR3), *C. grbavacensis* f. *lutea* (GRF8), *C. immanuelis-loewii* (IM), *C. melanocephala* (ME), *C. orientalis* (OR), and *C. zlatiborensis* (ZL); see Table S2 for population codes.

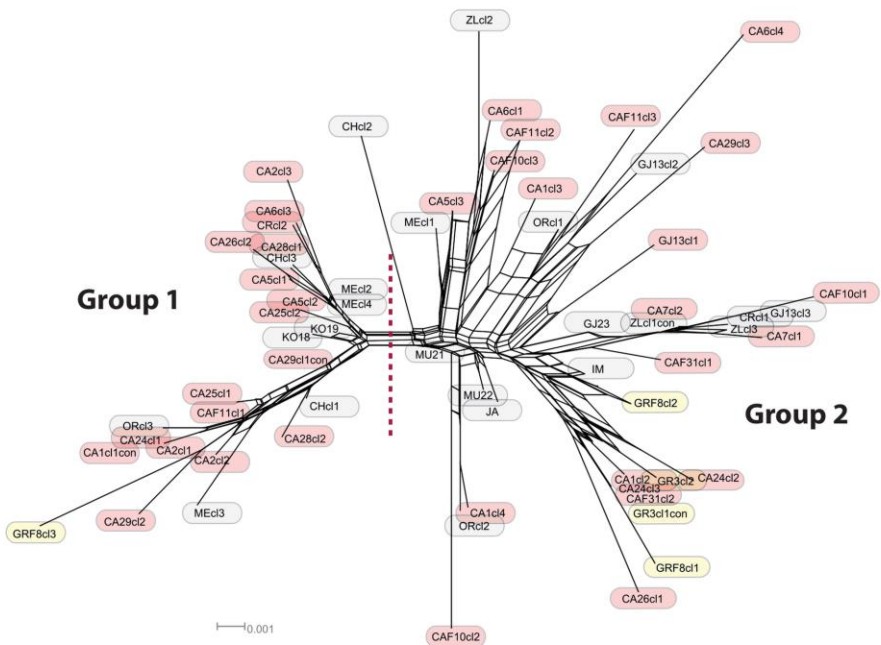

**Figure 5.** Neighbor-Net for the ETS region of the Centaurea calocephala complex. Population codes are detailed in Table S2. Abbreviations: cl = clon; con = consensus sequence. A dotted red line indicates the boundaries between the two groups.

Regarding the *AGT1* network, it also shows intense reticulation. As was the case in the ETS, most populations have sequences that are intermixed within the network (Figure 6).

*3.3. Chromosome Numbers*

All investigated populations were diploid with $x = 10$ or $x = 11$. Chromosome numbers of *C. gjurasinii* with $2n = 20$ and *C. melanocephala* with $2n = 22$ were determined for the first time. Chromosome numbers in eight populations of *C. calocephala* from different parts of the investigated area (Croatia, Romania, and Serbia) were in all cases $2n = 20$ (Table S2). For *C. kotschyana*, the chromosome number was $2n = 22$ in the investigated population from Serbia compared to $2n = 20$ and $2n = 22$ available in the literature (Table S2). For *C. murbeckii*, we counted $2n = 22$ in the population from Bosnia and Herzegovina differing from $2n = 20$ published by [86].

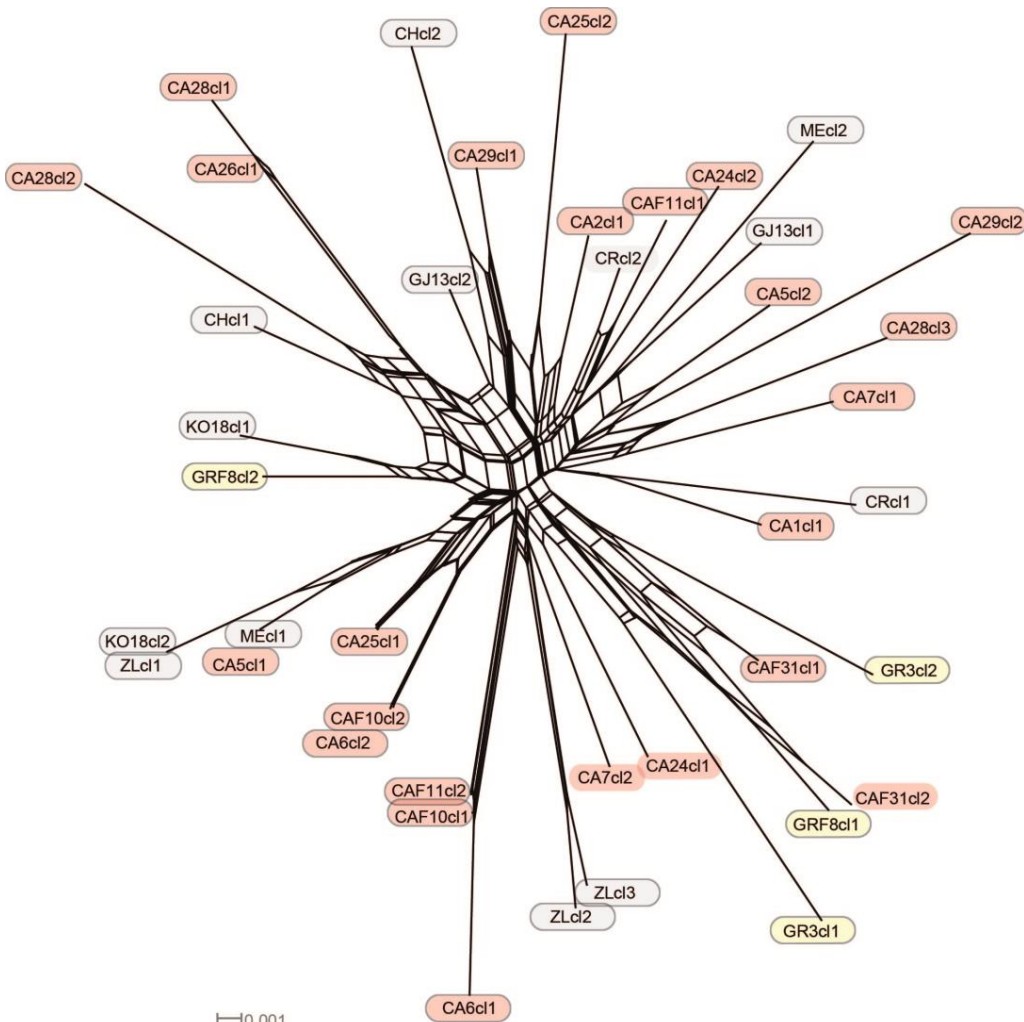

**Figure 6.** Neighbor-Net for the *AGT1* region of the Centaurea calocephala complex. Population codes are detailed in Table S2. Abbreviations: cl = clon; con = consensus sequence.

## 4. Discussion

### 4.1. Utility of Nuclear-Ribosomal ETS and Low-Copy AGT1 in Acrocentron

The backbone of the ETS tree supports the main geographical clades of sect. *Acrocentron* defined by [46] and confirms the monophyly of the Balkan + Eurasian clade (Figure 4). However, support is low within the major clades and (especially in the Balkan clade) some species show multiple copies placed in different subclades (Figure 4). Two reasons could explain the low resolution: recent speciation and introgression. Not many changes accumulate in sequences of fast-evolving spacers (ETS) in cases of fast speciations or when the species are very recently diverged; in our case, for such closely related species, the ETS gene tree had not yet attained reciprocal monophyly for the constituent species [87]. As for introgression, the ETS network is very illustrative, and ties are especially evident in Group 2 (Figure 5). However, given the multi-copy character of the rRNA-cistron, this conclusion should be taken with caution.

Low-copy *AGT1* showed fewer changes than the ETS, and the resulting phylogram was not informative (tree not shown). Ties were found to be especially intense in the network (Figure 6), to the extreme of blurring differences between groups. However, *AGT1* would help in suggesting the parentals of *C. zlatiborensis* better than the ETS, as discussed below.

*4.2. Phylogeny of the Balkan Clade: Introgression and Gene Flow*

Focusing on the results in the Balkan clade, *Centaurea salonitana* and *C. rupestris* are widely distributed in the south of the Balkans and are at the origin of the clade. In addition, at the base of the Balkan clade, we also find *C. ragusina*, an endemic Croatian species that inhabits crevices of limestone cliffs near the coast and islands of the Adriatic Sea. The rest of the Balkan clade is a polytomy with some well-supported subclades, which, however, do not correlate with current systematics. In fact, only *C. kotschyana* forms a well-supported subclade, while other subclades represent a mixture of morphologically and taxonomically well-defined species. For example, subclade B1 in Figure 4 is composed of cloned ribotypes from *C. calocephala*, *C. zlatiborensis*, *C. gjurasinii*, and *C. crnogorica* (Figure 1), while subclade B2 in Figure 4 is composed of cloned rybotypes of *C. calocephala*, *C. grbavacensis*, *C. melanocephala*, and *C. orientalis* from the Centaurea calocephala complex (Figures 1 and 2), and Iberian *C. cephalariifolia* and *C. legionis-septimae*, as well as an Italian population of *C. scabiosa*. Furthermore, most of the species of our study are placed in different subclades because they have more than one clone. In the ETS phylogeny, *C. calocephala* is placed in four different subclades; *C. orientalis* in three; and *C. cephalariifolia*, *C. crnogorica*, *C. gjurasinii*, *C. grbavacensis*, *C. melanocephala*, *C. scabiosa*, and *C. zlatiborensis* in two subclades (Figure 4). There are only two exceptions: individual CAF31 of *C. calocephala* var. *flava* shows only one clone close to *C. grbavacensis* clones, and population CA2 of *C. calocephala* shows only one clone (Figure 5). All this indicates that the morphologically well-delimited species from the Centaurea calocephala complex cannot be defined on a molecular basis, nor does the phylogeny of this group of species match the accepted systematics. This is not an isolated case in which molecular investigation thus far did not allow to a better separation of species. There are some other examples in the Balkans, such as *Knautia* [88], *Heliosperma* [89], and *Sempervivum* [90], where hybridization and reticulate evolution complicate matters when it comes to the field of systematics.

The fact that species of the Balkan clade have more than one clone is probably the result of hybridization events, even though other explanations are possible. If our hypothesis of the multiple clones were acquired by hybridization is true, this implies the presence of rampant ancient homoploid hybridization between species not currently sympatric (they are diploid with $2n = 20$ or $2n = 22$, Table S2). According to [51], altitude shifts caused by glaciations in the Iberian Peninsula allowed mutual contacts of taxa from the sect. *Acrocentron* currently located in the mountains, and this could also be the case in the *C. calocephala* complex. Glaciations could have had a major impact on the evolution and phylogeny of the group, because according to [91,92], *C.* sect. *Acrocentron* originated in the Pliocene. This must be the case in the Balkans, which were a refuge for many species during the Ice Age, as evidenced by the great species diversity. It is a well-known case for many plant groups that during glaciations, populations migrated into refugia and became temporarily sympatric [93]. For example, isolated populations of *Cardamine maritima* came into contact during the glaciation along the Balkan coast and in mountain massifs, which caused gene flow between species [94]. There are also similar hypotheses of hybridization during glaciations in species of *Fraxinus* [95], *Edraianthus* [28], or *Sesleria* [33].

The question of whether molecular methods may clarify the origin of hybrid taxa is exemplified in *C. zlatiborensis*. Morphology suggests that *C. zlatiborensis* is closely related to *C. calocephala* [47,48]. Both networks place clones of *C. zlatiborensis* in different positions: one clone is always placed close to *C. grbavacensis* and the other among the *C. calocephala* clones (Figures 5 and 6). The *AGT1* gene is very precise in pointing to *C. kotschyana* as one of the parental lineages of *C. zlatiborensis*. The other parental lineage would be *C. grbavacensis*. Present distributions of *C. kotchyana* and *C. grbavacensis* do not overlap, but they probably were in contact in a refugium during glaciations and likely gave rise to hybridogenic *C. zlatiborensis*. After glaciations, the range of the hybrid changed in relation to the parent species, and *C. zlatiborensis* probably colonized the Dinaric Alps in the western part of Serbia during the postglacial period.

During the field research, we found some "mixed" populations of *C. calocephala* with individuals with red (var. *atropurpurea*), yellow (var. *flava*), and intermediate red-yellow/yellow-red (var. *mixta*) capitula, such as the populations from Đerdap (CAF10, CAF11, and CA6) and Lika (CAF31 and CA7), as well as the populations of *C. grbavacensis* from Northern Macedonia (Suva Gora, GRF8, and GR3). These taxa are well-differentiated on a morphological basis, and some are intermediate (specifically var. *mixta*), with differences in size, shape, and position of involucral bracts, probably all of them formed by homoploid hybridization involving *C. calocephala* and an unknow species with yellow florets. Supporting this hypothesis, all these populations have a clone close to *C. grbavacensis* clones in the *AGT1* network (Figure 6).

A frequently asked question is where to place the boundary between hybrids and hybridogenic species. F1 generation hybrids can be defined as the offspring created by crossing individuals that belong to different populations, have different adaptive abilities, and their existence depends completely on the parent taxa [96]. According to the biological concept of species, hybrids should be considered "good" species if they are reproductively or geographically isolated [97]. We can find hybrid zones formed by F1 generation hybrids and individuals with different recombinations that are active sources of new evolutionary types and species [98]. The problem occurs in hybrid zones when hybrids stabilize and produce fertile offspring, but they are not completely reproductively isolated from their parents, i.e., they can interbreed with them. This is an obvious issue when dealing with homoploid hybrids. The main problem is the historical concept of species, which are treated as static entities [99] and not as a dynamic relationship between ancestors and descendants, i.e., an evolutionary line [100,101]. The question is whether hybrids should be considered hybrids only when they are completely reproductively isolated from their parents and produce fertile offspring. Hybrid zones can be very significant and provide a wealth of information about the status and levels of populations that are "on the way" to becoming new species [98].

## 5. Conclusions

(a) The Balkan–Eurasian clade of sect. *Acrocentron* is a natural group.
(b) The *C. calocephala* complex as currently defined is not a monophyletic group, given that *C. cephalariifolia*, *C. glehnii*, *C. legionis-septimae*, and *C. scabiosa* are nested between other species of the complex. All these species should be included in any future study of the complex. Species are well-delimited from a morphological point of view, and they are the result of allopatric speciation in the Balkan Peninsula. In these morphologically well-defined species, successive hybridization events accompanied by introgression and gene flow caused by latitudinal and altitudinal shifts in the transits from glacial to interglacial periods were superimposed.
(c) As a result of this introgression, any reconstruction of the species' boundaries using sequence data is problematic within the group, and the use of other, more resolving markers should be considered.

**Supplementary Materials:** The following supporting information can be downloaded at https://www.mdpi.com/article/10.3390/d14050394/s1, Table S1: General chorological and ecological characteristics of analyzed representatives of Centaurea calocephala complex from the Balkans. Table S2: List of investigated species, origin of the plant materials, with voucher numbers of herbarium specimens, chromosome numbers, and GenBank accession numbers [42,43,48,49,51,86,102–113].

**Author Contributions:** Conceptualization, N.G.-J., A.S. and P.J.; methodology, N.G.-J.; investigation, J.N., P.J., M.L., D.L., P.D.M., I.B., S.M., B.Z. and N.G.-J.; writing—original draft preparation, J.N.; writing—review and editing, A.S., N.G.-J., P.J., M.L., D.L., P.D.M., I.B., S.M. and B.Z.; funding acquisition, P.J., P.D.M., N.G.-J and A.S. All authors have read and agreed to the published version of the manuscript.

**Funding:** This research was funded by Serbian Ministry of Education, Science and Technological Development, grant no. 451-03-68/2020-14/200178, and the Catalan Government ("Ajuts a grups consolidats" 2017-SGR1116).

**Institutional Review Board Statement:** Not applicable.

**Informed Consent Statement:** Not applicable.

**Data Availability Statement:** Data other than DNA sequences are available from the authors by direct request.

**Conflicts of Interest:** The authors declare no conflict of interest.

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
