# Peer review of "Molecular Insights into the Centaurea Calocephala Complex (Compositae) from the Balkans—Does Phylogeny Match Systematics?"

_diversity, doi:10.3390/d14050394_

Round 1

Reviewer 1 Report

The manuscript presents a phylogenetic-evolutionary study on a group of Centaurea species with mainly Balkan distribution. The used marekers and methods are solid, but have their major shortcomings in a group like Centaurea (possibly a combination of recent speciation and reticulate evolution), as is quickly evident from teh manuscript. Still, some conclusions can be drawn from the data.

I have made comments in the manuscript and only want to emphasize the most importnat points here:

  • The Discussion has to be more balanced with respect to the causes potentially underlying the observed patterns; specifically, the evolution of rRNA loci needs to be fully considered (especially, since the number of loci is not known and everything is possible)
  • NeighbourNet is a fair starting point for addressing reticulate evolution, but it is more descriptive than anything; more sophisticated methods (like PhyloNet) could be attempted: it is well possible that incomplete lineage sorting cannot be rejected as explanation for the observed patterns. The division into two groups in the Networks is not evident from the networks themselves (you could assign two groups differently) and is rather a decision post-factum

Author Response

Dear Reviewer,

  Thank you for taking the time and making the effort to review our manuscript. Attached please find the new version of the paper. We have followed almost all the suggestions by both reviewers. I am attaching the PDF from reviewer 1 with our answers to his commentaries, a clean version, and a file with tracked changes.   Yours sincerely,

Jelica Novaković
Corresponding author

Reviewer 2 Report

The content of this article deserves to be published it includes a huge amount of caryological and molecular data. The choice to analyze only one sample per population is limiting, but it can no longer be corrected.

I hve marked minor corrections in the text.

Author Response

Dear Reviewer,

  Thank you for taking the time and making the effort to review our manuscript. Attached please find the new version of the paper. We have followed almost all the suggestions by both reviewers. I am attaching the PDF from reviewer 1 with our answers to his commentaries, a clean version, a file with tracked changes, and supplementary material corrected with the GB accession added. Just a note on reviewer 2: he has marked in the corrected file that the Centaurea calocephala complex should be in italics. However, when speaking of complex, the name should not be italicized.   Yours sincerely,

Jelica Novaković
Corresponding author

Round 2

Reviewer 1 Report

Only a few minor comments (line  numbers as in the uploaded manuscript) left:

l. 20: "confounds" instead of "compounds"

l. 37: either "in southeastern Europe" or "in the Southeast of Europe"

l. 57: "ca." instead of "cacistron"

l. 89-106: There is an issue with how these figures are embedded in the manuscript. Irrespective of that, there is no reason to show C. calocephala twice (especially, since these are the same pictures)

l. 130: "mountain plant" instead of "mountain plants" (remove s)

l. 130: according to https://www.researchgate.net/profile/Jelica-Novakovic-2/publication/334251523_Distribution_and_variability_of_Centaurea_kotschyana_Heuffel_ex_Koch_from_Central_Balkans/links/5d1f2fd792851cf44066955a/Distribution-and-variability-of-Centaurea-kotschyana-Heuffel-ex-Koch-from-Central-Balkans.pdf?origin=publication_detail C. kotschyana is also Carpathio-Balkanic, or do you use a different concept of C. kotschyana here?

l. 138: "arise" instead of "arises"

l. 225: "from a single accession" instead of "from an unique accession"

l. 256: "equipped" instead of "equiped"

l. 310: add period (.) at the end of the sentence

l. 327: "confounds" instead of "compounds"

l. 461: "allow a better" (remove "to")

l. 538: "likely gave rise to" instead of "likely originated"

l. 547: "on a morphological basis" (insert "a")

l. 548: "bracts. Probably, " (instead of "bracts, probably ")

Author Response

Dear Reviewer,

Attached please find the corrected version of the manuscript, we have followed all the suggestions. 

Yours sincerely, 
Jelica Novaković 
Corresponding author

This manuscript is a resubmission of an earlier submission. The following is a list of the peer review reports and author responses from that submission.